# Hypogonadism in Male Patients with Pituitary Adenoma and Its Related Mechanism: A Review of Literature

**DOI:** 10.3390/brainsci12060796

**Published:** 2022-06-17

**Authors:** Zisheng Yan, Ting Lei

**Affiliations:** Department of Neurosurgery, Tongji Hospital, Tongji Medical College, Huazhong University of Science and Technology, Wuhan 430030, China; d201981798@hust.edu.cn

**Keywords:** pituitary adenoma, hypothalamic–pituitary–gonadal axis, hypogonadism, testosterone

## Abstract

Maintaining normal gonadal axis hormone levels is important for improving the condition of male patients with pituitary adenoma. The current literature is somewhat divided on the results of evaluations of gonadal axis function in male patients with pituitary adenoma before and after treatment, and the increasing demand for better quality of life has provided motivation for this research to continue. In this article, we summarize the feasibility of using testosterone as an indicator for assessing male function and discuss the changes reported in various studies for gonadal hormones before and after treatment in male patients with pituitary adenoma. It is important for clinicians to understand the advantages of each treatment option and the effectiveness of assessing gonadal function. The rationale behind the theory that pituitary adenomas affect gonadal function and the criteria for evaluating pituitary–gonadal axis hormones should be explored in more depth.

## 1. Introduction

Pituitary adenomas are common benign tumors and the third-most-common primary intracranial tumors after meningiomas and gliomas [1,2]. Some of these tumors secrete excessive amounts of pituitary hormones, which cause endocrine metabolic disorders and damage to the corresponding target organs [3,4,5]. Pituitary adenomas can also compress normal pituitary structures and blood vessels, resulting in the secondary hypoplasia of the pituitary gland and involvement of the corresponding target glands [6,7]; these include the hypothalamic–pituitary–gonadal (HPG) axis, which is involved in male reproductive and sexual functions.

The HPG axis is crucial to the production of spermatozoa and the maintenance of sexual function in men. In men, the hypothalamus, pituitary gland, and testes form an integrated system that regulates both the proper synthesis and release of reproductive hormones and the production of healthy sperm [8]. Gonadotropin-releasing hormone (GnRH) is released intermittently by the hypothalamus and stimulates the pituitary gland to release pituitary gonadotropins, including pulsatile luteinizing hormone (LH) and follicle-stimulating hormone (FSH) [9]. LH and FSH are the main regulators of spermatogenesis; LH regulates testosterone production in mesenchymal cells (Leydig cells), and FSH synergizes with testosterone to regulate the production of the regulatory molecules and nutrients required for spermatogonia to develop into mature sperm cells [10]. Therefore, treatment that maintains the function of the HPG axis within a normal range is important for improving symptoms associated with pituitary adenomas in male patients.

Studies have shown mixed results regarding whether reduced serum testosterone levels indicate hypogonadism. According to current guidelines, the diagnosis of hypogonadism in adult men requires the presence of both low serum testosterone levels and at least one symptom of androgen deficiency; therefore, clinicians must attribute symptoms to hypotestosteronemia when making a diagnosis of hypogonadism. Due to the extensive overlap between symptoms associated with androgen deficiency and those of some common diseases, this may lead to the under- or over-diagnosis of hypogonadism [11,12]. While it is clearly controversial whether testosterone is useful as an indicator of male sexual function [13,14], it is plausible that reduced testosterone levels are an important risk factor for male sexual dysfunction [15,16].

There is a long history of research on gonadal hormones and their related sexual functions in patients with pituitary adenoma, and testosterone has received extensive attention as a driver of male sexual behavior.

## 2. Search Strategy

The following MeSH terms were searched for using PubMed and Web of Science: pituitary adenoma, hypogonadism, and testosterone. All reviews, case reports, and cohort studies published in English between 1975 and January 2022 were considered. Fewer articles were published before 1975 and often omitted important information, so they were not included. The exclusion criteria were as follows: (1) diagnosis of saddle lesions other than pituitary adenomas; (2) lack of more than three of the following seven characteristics: patient sex, age at diagnosis, clinical presentation, HPG axis-related hormone levels, diameter of the adenoma, treatment modality, and follow-up time; and (3) age under 18 years, because the reference range for pituitary adenoma incidence and gonadal hormone levels is different from that of adults. The diagnosis of secondary hypogonadism must be confirmed by hormone levels to be included in this review. We included 70 articles in this review, including 19 reviews, 2 case reports, and 49 cohort studies.

## 3. Pituitary Adenomas and Male Hypogonadism

In an analysis by Zhou et al. (2018), more than 50% of the 254 included male patients with pituitary adenomas had hypogonadism, the symptoms of which were very bothersome for these patients [6]. It has previously been found that, at the time of presentation, approximately 73–93% of patients with macroprolactinoma had evidence of FSH and LH deficiencies [17]. Fallah et al. [18] noted a rate of approximately 60% for gonadal hormone abnormalities in macroadenomas, while, in a study on non-functioning adenomas by Iglesias et al. [19], it was found that the gonadal axis was most often affected by pituitary adenomas, with a rate of 62.5% of abnormalities. The effects of various types of saddle lesions on male gonadal function have been described [20]. This shows that considering the gonadal hormone changes that occur in patients with pituitary adenomas and exploring the relevant therapeutic effects will have far-reaching implications for the treatment of male patients with pituitary adenoma and their quality of life.

Among patients with pituitary macroadenomas, the prevalence of hypogonadism is much higher in patients with functional pituitary adenomas than in patients with non-functional pituitary adenomas [21]. However, among patients with pituitary macroadenomas, whether functional or non-functional, most male patients suffer from significant sexual dysfunction. This suggests that, in patients with microadenomas or macroadenomas, the etiology of sexual dysfunction is primarily due to hormonal secretion, rather than a tumor-dominating physical effect, and that tumors of this size are less likely to cause sexual dysfunction in men by pressing on the pituitary gland. In patients with giant adenomas, the elevated proportion of sexual dysfunction may be the result of the tumor compressing the pituitary stalk, such that most patients with giant adenomas have sexual dysfunction, regardless of the pathological type of the tumor [6].

Many studies have reported that the most common etiologies of clinical hypogonadism are prolactinomas and gonadotropin adenomas [21,22,23]. Prolactinomas are the most common type of functional adenoma; hypoactive sexual desire or ED are found in 78–100% of male patients with large prolactinomas or microprolactinomas [24]. Sexual dysfunction is one of the most common symptoms of prolactinoma in men [21,25,26]. In the HPG axis, GnRH-stimulated prolactin (PRL) release is perhaps the first example describing the paracrine regulation of lactocytes [27]. In vitro, GnRH exerts a stimulatory effect on PRL secretion only when lactocytes are co-cultured with gonadotropins, suggesting that GnRH itself may not be responsible for stimulating PRL secretion, but may be a product of other gonadotropin-related factors [28]. PRL plays a role in regulating gonadotropin secretion, mainly in the form of acting on the hypothalamus to inhibit the synthesis and secretion of GnRH, and thereby decreases pituitary gonadotropin synthesis and secretion [29,30]. It also acts directly on the peripheral gonads to inhibit gonadotropin production [28]. In humans, hyperprolactinemia is associated with a significant reduction in the frequency and amplitude of the LH pulse, and the suppression of the LH pulse can be reversed by reducing the serum PRL concentrations to the normal range [31,32,33]. In a related experiment in male rats, PRL not only suppressed the frequency and amplitude of the LH pulse in males, but also induced the inhibition of GnRH release [34,35,36]. It has been shown that the inhibition of GnRH release by PRL is achieved through the concentrated inhibition of kisspeptin neurons in the hypothalamus. PRL receptors are expressed in most kisspeptin neurons [37]. The latest finding is that PRL induces the phosphorylation of signal transduction and activator of transcription 5 (pSTAT5) in the kisspeptin neurons of the arcuate nucleus in rats [38].

Patients with non-functioning pituitary adenomas can also have abnormal gonadotropin levels and may complain of symptoms related to sexual dysfunction at the time of presentation. Reports in the literature suggest that these symptoms are mainly due to non-functioning pituitary adenomas causing a pulling effect on the pituitary stalk or a compressive effect on the pituitary gland. On one hand, non-functioning pituitary adenomas may interrupt the hypothalamic–pituitary portal circulation. Subsequently, this interferes with the normal dopaminergic inhibition of PRL cells, resulting in elevated serum PRL levels. On the other hand, the direct pressure of the tumor on normal pituitary tissue may destroy this tissue. Furthermore, the disruption of portal circulation may induce the focal ischemic necrosis of normal pituitary cells. Due to the limited regenerative potential of the normal pituitary gland [39], the hypogonadism that results in some patients may be difficult to reverse [7]. In addition, the growth and development of non-functioning pituitary adenomas are often difficult to detect because they do not secrete clinical features with biologically active hormones [40]; this may lead to the delayed diagnosis of the disease in clinical practice. If not found incidentally, they are usually detected because of the above-mentioned symptoms associated with tumor growth. When a tumor compresses the pituitary gland [41,42], symptoms of partial or total hypopituitarism may occur as a result of the insufficient secretion of some or all of the pituitary hormones. In male patients, the most common pituitary dysfunction due to this cause is hypogonadism [43].

Other hormone-secreting pituitary adenomas may also cause hypogonadal symptoms in men; these include growth hormone adenomas, which are a category of functional pituitary adenomas. However, hypogonadism as the sole presentation in patients with acromegaly is extremely rare [21]. According to the Pituitary Tumor Registry database, 49% of male patients with acromegaly suffer from hypogonadism; of these patients, 45% have concurrent hyperprolactinemia. In patients with acromegaly, it has been suggested that hypogonadism may be caused by hyperprolactinemia, rather than growth hormone overload. Among men with growth hormone microadenomas, 39% had testosterone deficiency, but the majority still had serum PRL levels within the normal range. These data suggest that there may be other factors in addition to hyperprolactinemia that contribute to the pathogenesis of hypogonadism in men with acromegaly [44]. Lotti et al. found that male patients with acromegaly were at significantly increased risk of ED, especially those with cardiovascular disease; however, they found no significant correlation between ED and the serum growth hormone (GH) or testosterone levels. They concluded that the main cause of ED in patients with acromegaly is psychological, since abnormal growth hormone secretion can cause changes in the physical appearance of male patients, and chronic diseases can cause the loss of sexual desire in patients with growth hormone adenoma [45].

## 4. Changes in Male Gonadal Function after Treatment of Pituitary Adenoma

### 4.1. Effect of Surgical Treatment on Male Gonadal Function

It is now accepted that surgery is the preferred treatment modality for pituitary adenomas other than prolactinomas [46,47]. However, because studies have selected different types of pituitary adenomas for analysis, different effects of surgery on gonadal function have been derived (Table 1). A retrospective analysis of gonadal function in 109 patients with non-functioning pituitary adenomas in a 2017 study in Portugal revealed an approximately 60% prevalence of hypogonadism among the male patients, which is consistent with the 26–72.6% reported in other studies [48,49,50,51]. Among the male patients who underwent surgery, 14% experienced deterioration in gonadal function, 69% maintained a level similar to that before surgery, and only 17% of men had improved gonadal function. This follow-up result was from a first assessment of hormone levels measured 3–9 months after surgery [52]. In an earlier study, aging and hypogonadism were reported as independent risk factors for the development of ED [53].

In contrast, the percentage of men with sexual dysfunction before surgery was 62.6% in a study by Zhou et al. In addition, according to International Index of Erectile Function (IIEF-5) scores, giant pituitary adenomas were significantly smaller than pituitary microadenomas and macroadenomas. There was also a significant improvement in ED symptoms after surgical intervention in all male patients. The magnitude of this improvement was related to the degree of tumor resection; there was better relief of sexual dysfunction following the complete resection of the tumor than there was from partial resection. The remission of sexual dysfunction also varied for different hormone-producing types of tumors. There was a significantly higher prevalence of sexual dysfunction in men with prolactinomas, which improved significantly more after surgical intervention. Most importantly, this study suggested that, in Chinese men, a decrease in the preoperative testosterone levels was significantly and negatively correlated with an increased prevalence of sexual dysfunction; this suggests that the testosterone levels could be used as a sensitive indicator for predicting male sexual dysfunction [6].

In 2020, Baldia and his colleagues evaluated the postoperative hormonal outcomes of non-functioning pituitary macroadenomas and found a significant correlation between tumor volume and pituitary dysfunction prior to surgical intervention [57]. This result is similar to the findings reported by Jahangiri et al. [55] and Laws et al. [54]. The pituitary dysfunctions included gonadal dysfunction, thyrotropic dysfunction, and corticotropic dysfunction; the rate of dysfunction in these three axes varied depending on the tumor characteristics of the study population. For gonadal axis dysfunction, as judged by the hormone levels alone, the preoperative dysfunction rate was approximately 25%; studies by Fallah et al. [18] and Iglesias et al. [19] reported gonadal axis hormone abnormalities of 61.3% and 62.5%, respectively. After surgical interventions, more than 80% of the patients in these studies had their gonadal axis hormone levels restored to levels similar to those before surgery; however, the percentage of patients with worse function was significantly higher than the percentage of patients with elevated function [57]. The results of numerous studies suggest that there is no clear consensus on the effect that surgical intervention has on post-surgical pituitary adenoma gonadal function. The extent of surgical resection in patients included in the literature has not clearly defined whether there was total or subtotal resection, and most studies focused on decompression as the main aim of surgical intervention. The association between residual tumor size and postoperative gonadal function is not entirely certain, as there seems to be limited literature to date suggesting a specific size threshold for tumors affecting gonadal function. Perhaps a more convincing conclusion can be drawn by placing the resection rate in the context of total and subtotal resection.

### 4.2. Effect of Pharmacological Treatment on Male Gonadal Function

The main goal of treating patients with PRL is to restore PRL to normal levels while also normalizing the testosterone levels, shrinking tumors, and improving vision [21,24,58]. Since the use of dopamine agonists for prolactinomas was first proposed in 1978, pharmacological treatment has become the preferred modality for the treatment of most prolactinomas [25,58,59,60].

Dopamine agonists are effective in the treatment of most males with prolactinomas, restoring normal PRL levels in approximately 80–90% of men, regardless of tumor size changes, restoring normal testosterone levels in 60% of men, and reducing tumor volumes by more than 80% while preserving residual pituitary function [58,61]. The exact mechanism of the post-treatment remission of prolactinomas is not well studied; however, as is evident in follow-up results, most patients who present with sexual dysfunction note the improvement of their symptoms. With regard to the post-treatment restoration of gonadal function, it has been noted that dopamine agonists, such as cabergoline and bromocriptine, act not only by inhibiting PRL, but also at the central level to improve sexual function. This is common for dopamine receptor agonists; the use of apomorphine for ED also demonstrates this central-level effect [62].

In 2019, Shimon et al. [63] evaluated the effect of orally administered cabergoline on testosterone in older men and found that 15 of 21 men with reduced testosterone recovered to normal after a mean follow-up of 5.3 years. However, the follow-up period was not homogeneous in this study and the study endpoint was that most patients’ PRL levels returned to satisfactory levels with the drug; the study did not address the sensitivity of the drug in relation to the recovery of testosterone. In 2014, Tirosh et al. [64] studied the effect of cabergoline on hypogonadism in tumors of different sizes and found that the prevalence of hypogonadism was higher in patients with larger prolactinomas at presentation. The prevalence of hypogonadism was as high as 90.9% when the tumor size was greater than 40 mm, while the prevalence of hypogonadism in patients whose size was between 10 mm and 20 mm was 75.0%. The incidence of hypotestosteronemia was similar in men with prolactin adenomas of different sizes after drug treatment, with a rate of about 33.3%. However, the patients with larger prolactinomas in this study were treated with testosterone replacement therapy during follow-up; therefore, their post-treatment hypogonadism prevalence may be somewhat different from that of typical outcomes. In the study by Colao et al. [65], the pretreatment testosterone level was low in 30 of 41 male patients with macroadenomas (73.2%) and in 5 of 10 with microadenomas (50.0%). After 2 years of treatment with cabergoline, 25 patients with macroadenomas had normal testosterone levels (60.9%) and 6 of the patients with microadenomas had normal testosterone levels (60.0%), again yielding a similar proportion of normal testosterone levels. Some studies have reported different results. In 2020, Sehemby et al. [66] studied a group of hypogonadal patients in which the PRL levels were restored to the normal range after treatment with cabergoline; the LH–testosterone axis remained suppressed in these men. This phenomenon was also described in a 2014 study by Shimon and Benbassat [67]; after 3 years of follow-up, only 26.7% of patients were found to have recovered. Therefore, the authors concluded that greater baseline tumor sizes and serum PRL levels were predictors of persistent hypogonadism. Different tumor sizes may lead to differences in the testosterone recovery rates. In most cases, serum PRL levels correspond to the tumor size [7]. In an individual, changes in PRL may reflect a corresponding increase or decrease in tumor size. This also makes it difficult to distinguish the pathogenesis of a particular individual. Therefore, we believe that the effect of tumor size on gonadal function is more reliable when studied in pituitary adenomas without secretory function.

There are only a few studies that have described changes in gonadal function in patients who are not sensitive to pharmacological treatment [68]. In a study conducted by Shimon et al. in 11 men with hyperprolactinemia, there were cases in which PRL was not restored to within the normal range, but the testosterone levels returned to normal [67]. While this phenomenon is not uncommon in clinical work, the mechanisms involved are not clear at this time, and few studies have been reported in the literature to date.

Drug-sensitive macroprolactinomas can shrink rapidly with dopamine agonist therapy, leading to the rapid restoration of anterior pituitary function as well as the improvement or resolution of visual field defects. Indeed, according to the guidelines, visual field defects in macroprolactinomas are no longer considered a neurosurgical emergency, as pharmacological treatment has been shown to reverse these defects to a similar extent to surgical decompression [69]. However, the improvement of gonadal function in patients with poor drug treatment has rarely been reported; hence, it is of interest to explore the mechanisms by which pituitary tumors affect gonadal function. The definition of drug sensitivity criteria and the duration of medication follow-up vary considerably from study to study; therefore, the relevant conclusions they draw are often inconsistent. Perhaps by establishing uniform criteria and grouping different drug sensitivities in future studies, there may be a stronger indication to determine the mechanism by which these tumors affect gonadal function.

## 5. Conclusions

In conclusion, most of the existing studies that evaluated the therapeutic effects of pituitary adenoma treatments focused on the protection of visual acuity or the function of the cortisol and thyroid axis; however, there have been relatively few studies on the gonadal axis function, and there is considerable variation among the conclusions reached by the studies, including some disagreement regarding the evaluation criteria and therapeutic views [1,20,21,70,71]. Except for a few studies, the specific effects of PRL levels and tumor sizes have not been described. In addition, despite the improvement in surgical techniques and the accumulation of experience in the use of drugs, the effect of changes in treatment techniques on male gonadal function has been summarized in only a few studies. Therefore, further exploratory studies should be conducted in these areas.

## Figures and Tables

**Table 1 brainsci-12-00796-t001:** Overview of male gonadal function recovery after transsphenoidal surgery in the literature.

Author	Year	Patient/Gonadal Abnormalities	Type	Size	Surgical Method	Follow-Up	Recovery Rate
Nomikos [42]	2004	604/463	NFPA	All	Microscope	3 Months	15.9%
Laws [54]	2016	80/27	All	Macro	Endoscope	3–30 Months	≤7.4%
Jahangiri [55]	2016	198/40	NFPA	All	Microscope	1.5 Months	26%
Harary [56]	2018	160/56	NFPA	Macro	Endoscope	3–51 Months	21.3%
Fallah [18]	2019	79/48	All	Macro	Endoscope	3 Months	10.1%
Iglesias [19]	2021	40/31	NFPA	Giant	Both	6 Months	18.9%

NFPA: non-functioning pituitary adenoma.

## Data Availability

No new data were created or analyzed in this study. Data sharing is not applicable to this article.

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
