# Peer review of "Hypogonadism in Male Patients with Pituitary Adenoma and Its Related Mechanism: A Review of Literature"

_brainsci, 2022, doi:10.3390/brainsci12060796_

Round 1
Reviewer 1 Report
The review aims to describe male hypogonadism in patients with pituitary adenomas and focus on the leading mechanisms explaining this complication.
Despite the aim, the manuscript appears not well organized. Therefore, I suggest some modifications:
- Section 2 (Testosterone and Male Sexual Function) is not essential, and I suggest removing it. A brief introduction of male hypogonadism diagnosis may be moved up in the section "Introduction".
- More epidemiological details could be added to describe better the incidence and prevalence of male hypogonadism in patients with pituitary adenomas.
- Given the review aim, the mechanisms of male hypogonadism in pituitary adenomas should be discussed in more detail (compressive, mediated by hyperprolactinemia, due to excessive hormonal secretion).
- A more detailed presentation of the effects of medical management of pituitary adenomas could be included in section 4.2
Author Response
Comment 1: - Section 2 (Testosterone and Male Sexual Function) is not essential, and I suggest removing it. A brief introduction of male hypogonadism diagnosis may be moved up in the section "Introduction".
Response 1: First of all, thank you very much for your affirmation of our work. Our original intention in writing this section was to describe the relationship between testosterone and male sexual function, so that the later section on testosterone's association with the evaluation of efficacy would not be confusing. Based on your suggestion, we have kept the content related to diagnostic criteria in it and moved it to the introduction section. (Lines 41-50)
Comment 2: - More epidemiological details could be added to describe better the incidence and prevalence of male hypogonadism in patients with pituitary adenomas.
Response 2: Thank you very much for your advice on our work. Based on your prompting, we noticed that we did overlook this important element of the description before. We therefore added more descriptions of incidence and prevalence from previous literature to further highlight the importance of this study. (Lines 58-63)
Comment 3: - Given the review aim, the mechanisms of male hypogonadism in pituitary adenomas should be discussed in more detail (compressive, mediated by hyperprolactinemia, due to excessive hormonal secretion).
Response 3: Thanks very much for your suggestions. We are sorry that the previous version had too little review of the mechanics part, which is contrary to our original intention of writing. Based on your suggestion, we have added more details about the mechanisms in the corresponding section of Section 2 (previously Section 3): the mechanism of hormonal secretion causing hypogonadism has been added in lines 83-96, while the mechanism of compression factors causing hypogonadism has been added in lines 105-112.
Comment 4: - A more detailed presentation of the effects of medical management of pituitary adenomas could be included in section 4.2
Response 4: Thanks very much for your suggestions. In response to your prompt, we have added more detailed information on the pre-treatment condition in section 3.2 (previously section 4.2) (lines 213-217) and more data on remission after treatment in the subsequent sections. (Lines 219-224). These changes have made the article significantly more readable, and again, thank you for your endorsement and suggestions for this article.

Reviewer 2 Report
The authors present a data collection about the hypogonadism manifestation upon pituitary adenoma formation in males.
I think that the title not exactly fits with the issues presented in the review because the mechanism are not well described. In addition, the authors did not really discuss the results obtained by other studies rather providing only a description.
I suggest the authors to improve these aspects
some minor english spell check are required
Author Response
Comment 1: I think that the title not exactly fits with the issues presented in the review because the mechanism is not well described.
Response 1: Thank you very much for your advice on our work. We are sorry that the previous version had too little review of the mechanics part, which is contrary to our original intention of writing. Based on your suggestion, we have added more details about the mechanisms in the corresponding section of Section 2 (previously Section 3): the mechanism of hormonal secretion causing hypogonadism has been added in lines 83-96, while the mechanism of compression factors causing hypogonadism has been added in lines 105-112.
Comment 2: In addition, the authors did not really discuss the results obtained by other studies rather providing only a description.
Response 2: Thanks very much for your suggestions. I'm sorry that the previous version was not a good reading experience for you. Much of this was due to the lack of thought that went into the writing. We have therefore added the purpose of writing this article in the appropriate place (lines 181-187, lines 231-236, lines 243-255), and these are the things we are currently working on. We hope to summarize the content of the previous literature and do further research based on the establishment of new evaluation criteria.
Comment 3: some minor English spell check are required
Response 3: Thank you very much for the reminder. We have rechecked the entire text and sent it to an English editing company for a second edit, hoping to ensure your reading experience as much as possible.

Reviewer 3 Report
I think this is an interesting topic, however, my major objection to the manuscript is about methodological aspects.
I think I understand that it is a narrative review, however, this is not indicated in the title, nor is there any kind of section that talks about methodology.
I sincerely believe that this review would be much more useful if it were a systematic review, for which I recommend registering the review in a registry such as PROSPERO, and then I suggest using the PRISMA methodology to carry out the review.
It is very important to know the methodological quality and the risk of bias (GRADE and Pedro Scale for example) of the studies included in the study, it is also very important to be able to determine the level of evidence, conclusion, and recommendation after the review.
I advise you to redo the work taking into account all these recommendations.
Author Response
Comment 1: I think this is an interesting topic, however, my major objection to the manuscript is about methodological aspects.
I think I understand that it is a narrative review, however, this is not indicated in the title, nor is there any kind of section that talks about methodology.
I sincerely believe that this review would be much more useful if it were a systematic review, for which I recommend registering the review in a registry such as PROSPERO, and then I suggest using the PRISMA methodology to carry out the review.
It is very important to know the methodological quality and the risk of bias (GRADE and Pedro Scale for example) of the studies included in the study, it is also very important to be able to determine the level of evidence, conclusion, and recommendation after the review.
Response 1: First of all, thank you very much for your affirmation of our work. As you mentioned, the reading value of this article would indeed be greatly enhanced if it were written using a systematic review approach.
However, there have been fewer studies on gonadal function, and most of them have not been done on a large enough number of patients, due to previous medical constraints where people did not seek improvements in sexual function. Currently, as medical technology improves and patients' demands for improved gonadal function increase at the same time, more and more people will conduct studies in this area in the future. Therefore, these previous studies with small samples will have their corresponding reference value. We do not want to write in a standardized way, which may allow later researchers to overlook these results. After all, the studies we are currently conducting are also inspired by these studies with sample sizes of around 50 patients.
On the other hand, as mentioned in this article, there are differences in assessment criteria and treatment modalities across studies, and establishing workable uniform assessment criteria is itself one of the purposes of this writing.
We sincerely hope that you will understand this intention. Of course, if you insist that we change the way we write, we are willing to try to strengthen our training in this part and redo the work in subsequent revisions.

Round 2
Reviewer 3 Report
Thank you very much for your response and your willingness to improve the manuscript to make it more useful. Please introduce the aspects that I commented in the first revision.
I made my suggestions in the first revision and the authors have not followed them, however, they have indicated that if I insisted they would make them. This is the case I have insisted that they make the suggestions in the first revision.
In my opinion a narrative review can be very useful, but nowadays it is not acceptable under any scientific concept to accept a study that lacks methodology and therefore it is not possible to replicate it by another researcher. In this sense, since the authors should report in detail how they have carried out the literature search, in which databases, what inclusion and exclusion criteria they have followed, I have indicated that in order to be easily replicable they should follow the methodology of a systematic review, where the methodological quality of the articles included as well as the level of evidence, conclusion and recommendation are additionally reported.
In a literature review of any kind it is essential that there is a methodology section where all the aspects related to how the manuscript has been carried out are detailed so that another researcher can replicate it, this is not optional, it is essential, since this has to be done, it is more useful following a standardized guide.
The methodology followed in this review should include all the details of how it has been done are detailed, as well as the risk of sego and methodological quality of each of the articles, the level of evidence, conclusion and recommendation. The authors argue that there are hardly any studies and they are with a small sample, which is all the more reason to highlight this, given that the level of evidence is possibly very low and no type of recommendation can be made.
the section on methodology is missing seems to me to be very specific and that it is necessary to include it seems to me very specific, also after this comment I indicate a suggestion of how to do it so that it has the greatest possible scientific usefulness.
The authors have not followed a single one of my suggestions of the first review (they have not even changed the title), I believe that it cannot be found in any part of the text that it is a narrative review, is it a narrative review? what author of reference is used in the structure of this review? nowadays in a narrative review or any other type of review it is essential to indicate how it has been done.
Author Response
First of all, thank you very much for explaining the importance of structured writing, which has benefited us a lot and we fully realize the need for a separate paragraph on the methodology. Therefore, based on the advice you gave in the first round, we have rewritten and revised the following.
- Based on your suggestions, we have modified the title to reflect the type and character of the basis of our article. (Line 3)
- We added a new section 2: Search Strategy, which increases the credibility and reproducibility of the article by detailing the keyword search, engine type, literature time period, and inclusion exclusion criteria prior to writing this article, and brings the overall review structure closer to that of a systematic review. (Lines 25-66)
- Also, regarding the question, you pointed out about which review authors are cited in the article. Since the template given by the journal is based on a serial number search of the literature, in the individual citation sections we are using the serial number markers, but in the vast majority of the cited passages we mention the relevant authors of the article. For example, in line 69, when we describe the epidemiological data on pituitary tumor-associated hypogonadism, we not only mention the author Zhou et al, but also add the corresponding literature serial number in line 71. In this review, we cite authors including but not limited to some prominent neurosurgeons Colao A, Maiter D, Buchfelder M., et al.
- Finally, thank you again for your two rounds of advice, at which point we have a better understanding of review writing. However, we still need more time to learn how to systematically evaluate the quality of each evidence and the related recommendations. We are also acutely aware of this deficiency, and we will strengthen the relevant training in our subsequent writing according to your guidance.